# Effects of a Novel Cold Atmospheric Plasma Treatment of Titanium on the Proliferation and Adhesion Behavior of Fibroblasts

**DOI:** 10.3390/ijms23010420

**Published:** 2021-12-31

**Authors:** Ming Yan, Philip Hartjen, Martin Gosau, Tobias Vollkommer, Audrey Laure Céline Grust, Sandra Fuest, Lan Kluwe, Simon Burg, Ralf Smeets, Anders Henningsen

**Affiliations:** 1Department of Oral and Maxillofacial Surgery, University Medical Center Hamburg-Eppendorf, 20246 Hamburg, Germany; p.hartjen@uke.de (P.H.); m.gosau@uke.de (M.G.); t.vollkommer@uke.de (T.V.); a.grust@uke.de (A.L.C.G.); lkluwe@uke.de (L.K.); s.burg@uke.de (S.B.); r.smeets@uke.de (R.S.); a.henningsen@uke.de (A.H.); 2Department of Oral and Maxillofacial Surgery, Division of Regenerative Orofacial Medicine, University Medical Center Hamburg-Eppendorf, 20246 Hamburg, Germany; s.fuest@uke.de

**Keywords:** cold atmospheric plasma, osteoblast-like cells, titanium, proliferation

## Abstract

Cold plasma treatment increases the hydrophilicity of the surfaces of implants and may enhance their integration with the surrounding tissues. The implaPrep prototype device from Relyon Plasma generates cold atmospheric plasma via dielectric barrier discharge (DBD). In this study, titanium surfaces were treated with the implaPrep device for 20 s and assessed as a cell culture surface for fibroblasts. One day after seeding, significantly more cells were counted on the surfaces treated with cold plasma than on the untreated control titanium surface. Additionally, the viability assay revealed significantly higher viability on the treated surfaces. Morphological observation of the cells showed certain differences between the treated and untreated titanium surfaces. While conventional plasma devices require compressed gas, such as oxygen or argon, the implaPrep device uses atmospheric air as the gas source. It is, therefore, compact in size and simple to handle, and may provide a safe and convenient tool for treating the surfaces of dental implants, which may further improve the implantation outcome.

## 1. Introduction

The failure of implants to achieve good osseointegration within the surrounding bone tissue after implantation is often an important reason for the failure of implant surgery [1]. It has always been the research direction for the majority of scholars to improve the success rate of implantation by changing the surface properties of the implant, increasing the biocompatibility between the implant and the surrounding tissue, and reducing the risk of infection after implantation [2,3,4,5,6]. However, the success of implants requires not only good osseointegration but also a good soft tissue seal [7]. A good soft tissue seal can reduce the influence of microorganisms on the implant and protect the osseointegration [8]. The soft tissue around the implant is composed of epithelial tissue consisting of epithelial cells and connective tissue consisting of fibroblasts. It was found that epithelial cells have migrate along the surfaces of the implant towards the root, while healthy connective tissue can prevent the migration of epithelial cells towards the root. A good bond between connective tissue and the implant means that fibroblast can adhere and proliferate well on the implant surface. The factors that influence the cell adhesion and proliferation are the surface roughness, surface morphology, surface chemistry, and hydrophilicity [9]. Plasma is a term used to describe partially ionized gases, which can be classified into thermal plasma and low-temperature plasma. Thermal plasma can be used as a surface coating technology to spray metals and ceramics onto other materials. Metallic particles such as titanium, silver, and hydroxyapatite are treated using thermal plasma spraying technology to enhance the biocompatibility of dental or orthopedic implants [10]. In contrast, low-temperature plasma has a lower temperature and can be used to directly treat living tissues to achieve sterilization, blood coagulation, wound healing, and tissue regeneration [11]. It can also be used to indirectly treat living tissues by means of implanting plasma-treated materials [12,13]. In our previous work, the surfaces of titanium specimens were treated with an argon–oxygen plasma jet for 60 s [14,15]. It was found that plasma treatment can increase the wettability of the surfaces of titanium sheets and can also promote the early attachment of gingival fibroblasts on the surfaces of titanium specimens, which may enhance the sealing effect between the implant and the surrounding soft tissue, reduce the risks of infection by pathogenic bacteria, and enhance the ability of the implant to integrate within the surrounding bone tissues. The traditional plasma treatment systems (including vacuum devices and gas storage devices) are large in size, use high external voltages, and require external working gas, which makes them extremely inconvenient for clinical applications. With the development of industrial technology, the atmospheric plasma devices have the advantages of being small in size and light in weight and can directly use air as the working gas, which is convenient for clinical applications. In this study, the implaPrep prototype device from Relyon Plasma (Figure 1) was used to activate the surfaces of the materials, and the effects of the activation on the adhesion and proliferation of fibroblasts were observed to explore whether atmospheric plasma can improve the biological properties of bone implant materials, providing a reference for clinical applications.

## 2. Results

### 2.1. Water Immersion Angle

Implants that were plasma-treated for 20 s showed increased wettability with regard to the immersion angle measurement compared to the untreated reference (immersion angle 120° vs. 63°) (Figure 2).

### 2.2. Early Cell Adhesion

Figure 3 shows the early adhesion of L929 cells on the surfaces of both materials. Most of the cells adhered to the surfaces of the materials in a spherical shape and pseudopodia form, but there was no obvious spreading in the control groups. However, most of the cells were in a spread state, polygonal, with thick and long pseudopodia extending around them in the experimental groups (Figure 3).

### 2.3. Cell Proliferation

The proliferation of L929 cells was determined using an XTT assay. As shown in Figure 4, the numbers of cells in the two groups of materials gradually increased over time. The numbers of cells in the experimental groups were higher than in the control groups at 24, 48, and 72 h, and the differences were statistically significant (*p* < 0.05) (Figure 4). There were no statistical differences among the experimental groups.

### 2.4. Cell Count

The cell count results for the two treatments on the surfaces of the titanium specimens at different time points are presented as means ± standard deviations in Figure 5. Here, (x^−^ ± s) depicts the cell growth trend based on the number of cells on the titanium specimens. The results showed that the cells were able to adhere to the surfaces of the titanium specimens in each experimental group within 24 h of cell culture but grew slowly, while the number of cells in each group did not change much, which was considered the incubation and adaptation period of the cells. The numbers of cells in the experimental groups were higher than in the control groups at 24, 48, and 72 h, and the differences were statistically significant (*p* < 0.05) (Figure 5).

### 2.5. Assessment of Cytotoxicity

As expected, cytotoxicity was not detected on any of the surfaces, regardless of the treatment (Figure 6).

## 3. Discussion

The advantageous mechanical and biological properties of titanium have gained increasing attention from dental professionals. With continuous improvements in dental implant technology, titanium has good application prospects as a dental material [16]. However, titanium and its alloys are biologically inert materials, which can only form mechanical bonds with growing bone, not chemical bonds [17]. Therefore, it is necessary to perform a surface treatment on titanium and its alloys to promote biological activities, such as enhanced adhesion, proliferation, protein synthesis, contact angles, and matrix mineralization of osteoblasts on its surfaces, which could accelerate osseointegration. The establishment of osseointegration is the key to implant success.

The determination of the water immersion angle is a simple method used to measure the hydrophilicity of surfaces [18]. As can be seen from Figure 2, the plasma treatment resulted in increased wettability of the titanium surface, as evidenced by a reduced immersion angle. This increased wettability likely promotes cellular attachment.

During the process of osseointegration, cell adhesion and proliferation on the surface of the implant are prerequisites [19,20,21]. Therefore, when evaluating the biological activity of implant materials, the material’s ability to promote the early adhesion of osteoblasts on the surface and the formation of good cell morphology should be observed as important indicators. Moreover, the adhesion of cells to materials is the foundation of tissue engineering research. The adhesion and aggregation of cells on the surface of the implant plays a key role in contact osteogenesis [22]. The experiments showed that surface-coarsening treatment of the implants could promote the adhesion and proliferation of osteoblasts. Within 24 h in culture, mouse fibroblasts had attached well to the surfaces of titanium sheets in each treatment group, although there was no significant change in the number of fibroblasts. However, after 24 h in culture, the number of fibroblasts began to increase linearly, and the number of fibroblasts in the experimental group was greater than that in the control group at each time point. In addition, the fibroblasts on the surfaces of titanium specimens in each treatment group were well stretched and showed a prolonged-spindle shape, while the antennae and pseudopodia were not obvious in the control group and the number of adhered cells was also smaller. The cold plasma (traditional and atmospheric) and UV treatment groups were superior to the untreated group in this regard.

Prior to the clinical application of any biomaterial, a biological evaluation should be conducted in accordance with the relevant biosafety inspection criteria, including three stages of primary toxicity screening, animal experiments, and clinical trials [23,24]. The primary rapid toxicity screening program is divided into two systems: in vitro and in vivo. In vitro cytotoxicity testing is one of the important aspects. At present, the lactate dehydrogenase release assay is used to test the cytotoxicity of plant materials. This method has the advantage of sensitivity, simplicity, and objective quantification. In recent years, it has been gradually adopted to evaluate the cytotoxicity of biological and dental materials [25]. Lactate dehydrogenase (LDH) in culture medium was used as an indicator in this study [26]. Therefore, the cytotoxicity test was used in this experiment to study the titanium sheets treated by different methods. The experimental results showed that mouse fibroblasts could adhere to the surfaces of titanium sheets in each treatment group. There were no statistically significant differences among the groups (*p* > 0.05).

In vivo experiments should become an indispensable part of the soft tissue exploration process, especially because in vitro experiments cannot simulate the peri-implant fiber pathway. Further studies with in vivo experiments will be summarized in our next paper.

Currently, there is no relevant standard for plasma devices in clinical applications. Factors such as the gas type, treatment time, generated power, and exposure methods should be considered during application, because different conditions may have different biological effects. In this experiment, the implaPrep device developed by Relyon Plasma was applied to change the use of compressed gas to pure air, which can provide a more uniform, safe, and convenient medical treatment method. We believe that the application of atmospheric plasma in the field of stomatology has bright prospects.

## 4. Materials and Methods

### 4.1. Measurement of the Water Immersion Angle before and after Plasma Treatment

Dental implants with acid-etched and sandblasted titanium surfaces were analyzed regarding the water immersion angles before and after plasma treatment for 20 s. The immersion angle measurements were performed 30 min after plasma treatment.

### 4.2. Experimental Groups and Material Surface Activation Treatment

Two kinds of titanium material (acid-etched and sandblasted titanium and untreated titanium) were used in this experiment. Samples were randomly divided into one group of non-treated samples (controls) and three experimental groups, with three samples in each group. The materials were immersed in isopropanol and dried for disinfection. The disks in the novel plasma group were treated using the implaPrep prototype device for 20 s (relyon plasma is based on a cylindrical dielectric barrier discharge with an inner diameter of approximately 1.5 cm, powered by alternating current were adjusted to 24 V and 1.5 A) (Relyon Plasma GmbH, Regensburg, Germany). Disks in the traditional plasma group were treated with argon plasma for 60 s, using an NTP reactor (pressure 1 mbar, gas flow rate 1.25 sccm, and gas purity > 99.5%) (Diener Electronic GmbH, Ebhausen, Germany). Disks in the UV light experimental group were treated using a UV light oven for 12 min, which generated UV light with an intensity of 0.15 mW/cm^2^ (λ = 253.7 nm). Both of the low-temperature plasma systems are shown in Figure 1.

### 4.3. Cell Culture

L929 murine fibroblast cells (Sigma–Aldrich, Munich, Germany) were cultured in endothelial cell medium (ECM) containing 5% fetal bovine serum, 1% endothelial growth factor, and 1% penicillin/streptomycin at 37 °C and 5% CO_2_. The cell growth was observed and the medium was changed once every two days.

### 4.4. Cell Adhesion

Fluorescence-labeled phalloidin (biotinylated phalloidin, Alexa Fluor 488 green, 1:1000; Thermo Fisher Scientific, Waltham, MA, USA) and DAPl were used to label cytoskeleton filaments and nuclei, respectively. The following specific steps of staining were performed. The culture medium in the Petri dish was aspirated and then cells were washed with sterile PBS 3 times for 5 min and fixed with paraformaldehyde (40 g/L) for 30 min. The cells were washed again with sterile PBS three times, then 0.3% tritonX-100 solution (Gibco, Invitrogen, Paisley, UK) was added and the cells were incubated for 5 min to increase their permeability. The TritonX-100 solution was removed and DAPI was added to stain the cells for 15 min, then they washed with PBS three times. Phalloidin was added to stain the cells for 30 min in the dark, then they were washed with sterile PBS three times. Finally, the stained cells were visualized under a confocal laser scanning microscope.

### 4.5. Cell Proliferation Assay

Here, 1 mL of cell suspension with a density of 8 × 10^4^ cells per mL was seeded on the surfaces of the materials in experimental groups and control groups in 12-well plates. The cell growth on the surface of the material was observed under a microscope at 8, 24, 48, and 72 h, respectively. Next, 500 μL of XTT labeling solution was added to each well at 24, 48, and 72 h, which was further cultured at 37 °C for 4 h. Then, the 12-well plates were shaken for 1 min. Next, 100 μL of the supernatant from each well was transferred into a 96-well plate and three parallels were set for every sample. The OD values of the supernatants were measured at 450 nm and a reference wavelength of 650 nm.

### 4.6. Lactate Dehydrogenase Release Assay

Here, 10 ul of supernatant from each well was collected at 24, 48, and 72 h in culture. Then, LDH (LDH Cytotoxicity Assay Kit II, BioVision, Milpitas, CA, USA) reaction mix solution was added to the corresponding well of the 96-well plate and incubated in the dark for 30 min at room temperature. The OD values were measured at 450 and 650 nm (reference).

### 4.7. Observation of Cell Growth with Live–Dead Staining

The cells were washed twice with sterile PBS and then live–dead cell fluorescent dye (containing 100 μg/mL FDA and 60 μg/mL PI) was added to cells, which were then incubated in the dark for 3 min. The cells were washed twice again with sterile PBS and then cell viability was observed under a microscope. Five regions of interest were randomly selected and photographed and the numbers of cells were determined using Image J software.

### 4.8. Statistical Analysis

All experiments were repeated three times independently in duplicate wells. Statistical analysis was performed using SPSS 21 (IBM, Armonk, NY, USA). The significance of differences in viability and toxicity was assessed using *t*-tests. For all results, significant differences were defined as *p* < 0.05.

## 5. Conclusions

In this study, titanium was activated by atmospheric low-temperature plasma and the results showed that the activated titanium has promoting effects on cell adhesion and proliferation, and at the same time has no adverse effects on cytocompatibility. In the next step, our group will further study the role of titanium activated by atmospheric low-temperature plasma on enhancing osteogenesis.

## Figures and Tables

**Figure 1 ijms-23-00420-f001:**
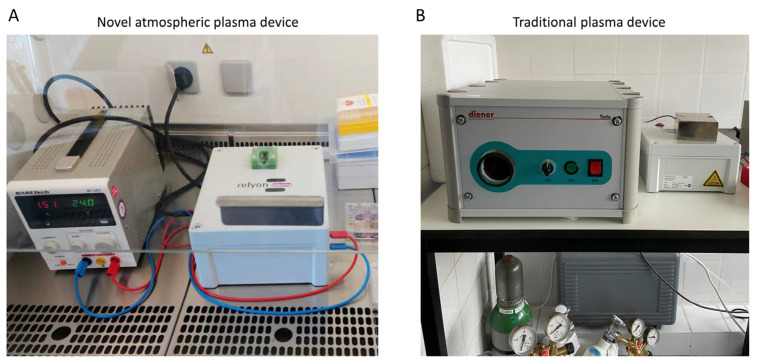
(**A**) The implaPrep prototype device from Relyon Plasma is based on a cylindrical dielectric barrier discharge system with an inner diameter measuring approximately 1.5 cm, powered by an alternating current. The equipment parameters were adjusted to 24 V and 1.5 A. (**B**) Traditional device reactor (generator frequency 100 kHz, input power 24 W, system pressure 1 mbar, gas flow rate 1.25 sccm, and gas purity >99.5%).

**Figure 2 ijms-23-00420-f002:**
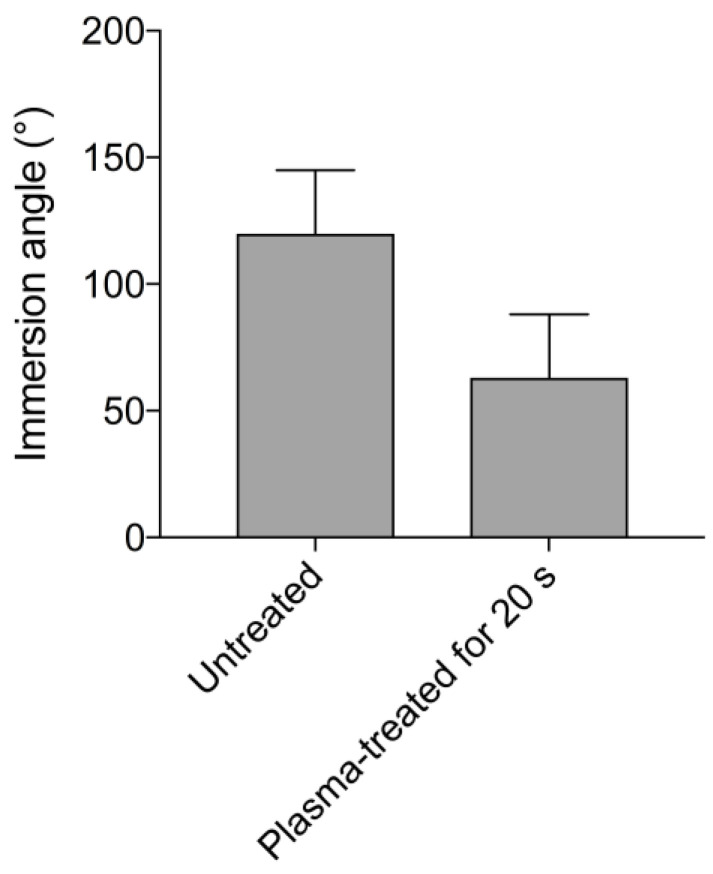
Dental implants with acid-etched and sandblasted titanium surfaces were analyzed regarding the water immersion angle before and after plasma treatment for 20 s. The columns represent mean values, while the error bars represent the standard deviation.

**Figure 3 ijms-23-00420-f003:**
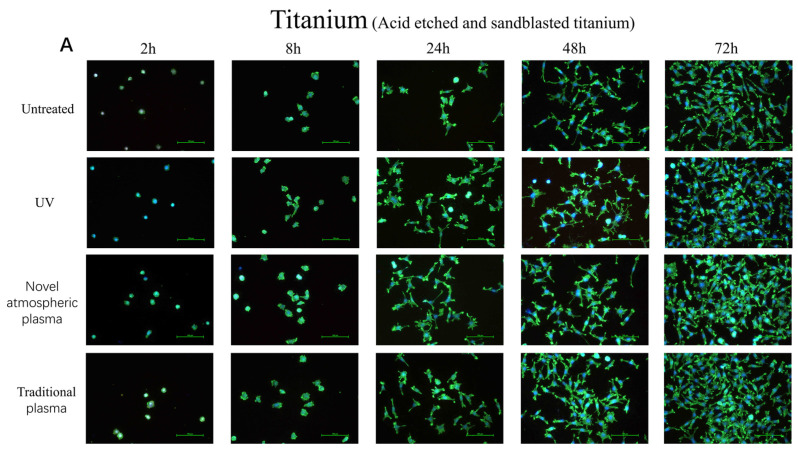
(**A**) Acid-etched and sandblasted titanium surface. (**B**) Untreated titanium surface. Cell attachment and morphology results for different surface-treated titanium specimens after 2, 8, 24, 48, and 72 h. Cells were cytoskeleton-stained with phalloidin. The cells increased in volume and adhered to the surfaces. The morphology of the cells was more extended and the cells were larger, especially on plasma- and UV-light-treated surfaces. In the plasma- and UV-light-treated groups, the cells were most extended and showed more filopodia-like structures.

**Figure 4 ijms-23-00420-f004:**
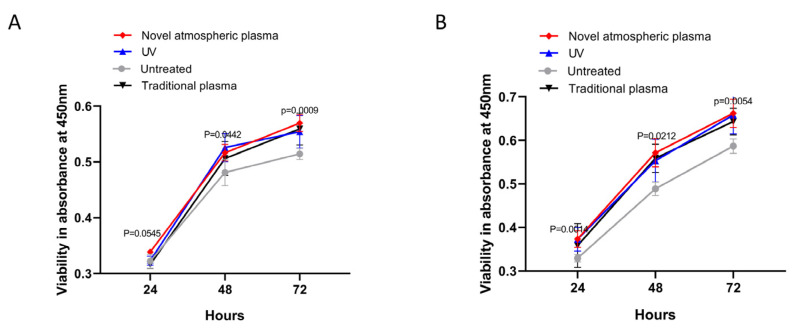
(**A**) Acid-etched and sandblasted titanium surface. (**B**) Untreated titanium surface. The cell counts (proliferation) were analyzed using XTT at 24, 48, and 72 h.

**Figure 5 ijms-23-00420-f005:**
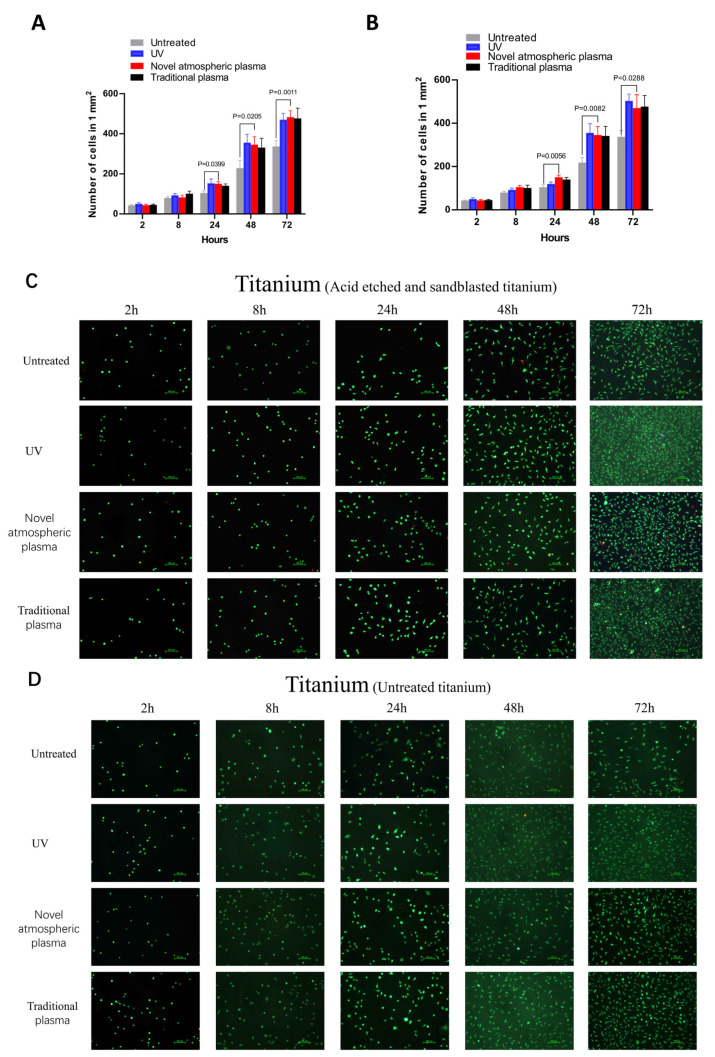
(**A**,**B**) The cell counts (proliferation) were analyzed using ImageJ software at 2, 8, 24, 48, and 72 h. (**C**,**D**) The morphologies of the fibroblasts were determined at 2, 8, 24, 48, and 72 h by direct observation with a light microscope after live–dead staining. Scale bar: 100 μm. (**A**,**C**) Acid-etched and sandblasted titanium surface. (**B**,**D**) Untreated titanium surface.

**Figure 6 ijms-23-00420-f006:**
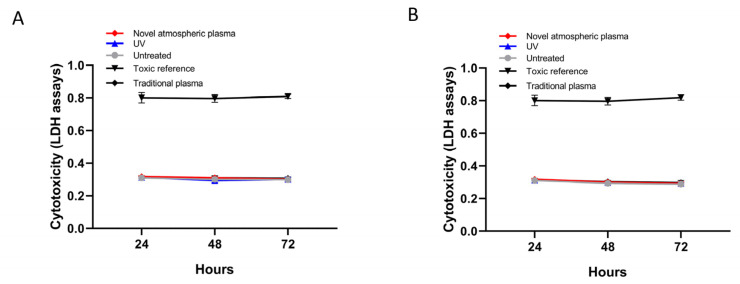
Cytotoxicity of different surface-treated titanium specimens after 24, 48, and 72 h, measured using an LDH assay. (**A**) Acid-etched and sandblasted titanium surface. (**B**) Untreated titanium surface. No significant differences were detected.

## Data Availability

Not applicable.

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
