# Peer review of "Effects of a Novel Cold Atmospheric Plasma Treatment of Titanium on the Proliferation and Adhesion Behavior of Fibroblasts"

_ijms, 2021, doi:10.3390/ijms23010420_

Round 1

Reviewer 1 Report

This study suggests the improved cell activity on Ti implant materials treated athmospheric cold plasma spray method. The results showed promising enhancement in terms of cell activities. However, as the authors indicated in the introduction, these phenomena depend on the surface characteristics of the samples. Thus, I suggest to the authors consider the change in surface characteristics before and after cold plasma spraying, such as topology, water contact angle, and so on. This study still requires further investigation on surface characteristics. Therefore, I recommend major revision with the addition of surface analysis and rewriting related parts for revised submission.

Author Response

This study suggests the improved cell activity on Ti implant materials treated athmospheric cold plasma spray method. The results showed promising enhancement in terms of cell activities. However, as the authors indicated in the introduction, these phenomena depend on the surface characteristics of the samples. Thus, I suggest to the authors consider the change in surface characteristics before and after cold plasma spraying, such as topology, water contact angle, and so on.

Replay: We thank the reviewer for this helpful comment. We agree that a comparison of the surface characteristics before and after cold plasma treatment would improve our manuscript and thus we added a water immersion angle analysis of the titanium surface before and after plasma treatment, which demonstrates an increased wetability of the plasma-treated surface.

This study still requires further investigation on surface characteristics. Therefore, I recommend major revision with the addition of surface analysis and rewriting related parts for revised submission.

Reply: We added a figure with the recommended surface analysis and rewrote the related passages of the manuscript.

Reviewer 2 Report

Dear authors, 

the article is interesting for the actuality of the topic and the possible clinical consequences. However, a major revision of the article is suggested in order to improve your work and make it more readable to other researchers.

In the introduction part, a section dedicated to implant survival rate should be inserted. Different implants and different tissues (based on the quantity of keratinized gingiva) in the oral cavity are related to different survival rate. In this way, a clearer connection between the role of fibroblasts and implant survival can be introduced.

In your study a group of titanium treated with traditional plasma system is missing. Why don't you consider it?
In the discussion a focus on the differences between these two systems is needed and eventually included in the limits of the study.

In the introduction, I kindly suggest proper citations of work of Canullo et al. that is active in this field. 

In the results section, it is not declared if there are any statistical differences or not between the two experimental groups. Please clarify

Line 154-155: "The results indicated that the titanium sheets treated by cold plasma and UV had better cytocompatibility than those in the untreated group". This phrase is incorrect. There are no better or worse results. The results can be equivalent or statistically different. Erase the phrase or change it.

In materials and methods section (line 176-178), you wrote that "two kinds of titanium materials including Promote-Oberfläche and glatt gebeitzt were used. What are the differences between these two types of surfaces? Why you take these two? Please explain

Line 231-232: Titanium has been widely used in ... guiding tissue regeneration. It's not proper like this. Titanium is not a biomaterial for guided tissue regeneration. Titanium can osseointegrate thanks to its properties but it cannot regenerate hard or soft tissues.

A section of limits of the study is lacking. Please write it at the end of the discussion 

Author Response

Response to Reviewer 2 Comments

Dear authors,

the article is interesting for the actuality of the topic and the possible clinical consequences. However, a major revision of the article is suggested in order to improve your work and make it more readable to other researchers.

In the introduction part, a section dedicated to implant survival rate should be inserted. Different implants and different tissues (based on the quantity of keratinized gingiva) in the oral cavity are related to different survival rate. In this way, a clearer connection between the role of fibroblasts and implant survival can be introduced.

Reply: We thank the reviewer for this suggestion. We revised the manuscript accordingly.

In your study a group of titanium treated with traditional plasma system is missing. Why don't you consider it?
In the discussion a focus on the differences between these two systems is needed and eventually included in the limits of the study.

reply: We fully agree and added data that was generated using a traditional plasma system.

In the introduction, I kindly suggest proper citations of work of Canullo et al. that is active in this field.

reply: We revised the introduction section accordingly.

In the results section, it is not declared if there are any statistical differences or not between the two experimental groups. Please clarify

reply: There is no statistical difference between the two experimental groups. We revised the results section to clarify this.

Line 154-155: "The results indicated that the titanium sheets treated by cold plasma and UV had better cytocompatibility than those in the untreated group". This phrase is incorrect. There are no better or worse results. The results can be equivalent or statistically different. Erase the phrase or change it.

reply: We thank the reviewer. We deleted this phrase as suggested.

In materials and methods section (line 176-178), you wrote that "two kinds of titanium materials including Promote- Oberfläche and glatt gebeitzt were used. What are the differences between these two types of surfaces? Why you take these two? Please explain

reply: We thank the reviewer for this comment. we neow use the following names for the materials: "Untreated titanium" and "Acid etched and sandblasted titanium"

Line 231-232: Titanium has been widely used in ... guiding tissue regeneration. It's not proper like this. Titanium is not a biomaterial for guided tissue regeneration. Titanium can osseointegrate thanks to its properties but it cannot regenerate hard or soft tissues.

reply: We deleted the phrase as suggested by the reviewer

A section of limits of the study is lacking. Please write it at the end of the discussion

reply: We added a passage about the limitations of the study as suggested by the reviewer.

Round 2

Reviewer 1 Report

Dear Author,

It is unfortunate to say that I am unable to see the figures in the revised version of your manuscript. Thus, I will request revision. This problem is most likely related to the pdf preparation process of the journal system.

Reviewer 2 Report

Thanks for your corrections and congratulations for your work

Round 3

Reviewer 1 Report

I believe the manuscript is suitable for publication after the revision.